# miRNAs in Lung Development and Diseases

**DOI:** 10.3390/ijms21082765

**Published:** 2020-04-16

**Authors:** Eistine Boateng, Susanne Krauss-Etschmann

**Affiliations:** 1Early Life Origins of Chronic Lung Diseases, Research Center Borstel, Leibniz Lung Center, Member of the German Center for Lung Research (DZL), 23845 Borstel, Germany; skrauss-etschmann@fz-borstel.de; 2Institute for Experimental Medicine, Christian-Albrechts-Universität zu Kiel, 24118 Kiel, Germany

**Keywords:** lung development, lung diseases, miRNA, human, mouse

## Abstract

The development of the lung involves a diverse group of molecules that regulate cellular processes, organ formation, and maturation. The various stages of lung development are marked by accumulation of small RNAs that promote or repress underlying mechanisms, depending on the physiological environment *in utero* and postnatally. To some extent, the pathogenesis of various lung diseases is regulated by small RNAs. In this review, we discussed miRNAs regulation of lung development and diseases, that is, COPD, asthma, pulmonary fibrosis, and pulmonary arterial hypertension, and also highlighted possible connotations for human lung health.

## 1. Introduction

The direct interaction between the environment and the mammalian respiratory structures plays a critical role in the stimulation of several pathological conditions that disrupt lung function [1,2,3]. Moreover, molecular alterations in the embryonic lung [4,5] are detrimental to normal organ formation. Advances in scientific methods have so far helped explain some of the complex signaling networks in the developing lung [6,7] and also explored pulmonary stem cells for the purpose of tissue regeneration in lung diseases [8]. The functions of small non-coding RNAs in cell differentiation and organ regeneration [9,10] also underscore their value in lung development and diseases.

The developing mammalian lung undergoes a series of sophisticated stages [11], and the expression of specific miRNAs at each level *in utero* and extra-uterine life may influence organ morphogenesis, maturation, and pathogenesis of diseases. The role of miRNAs in lung development became evident when *Dicer1*, a gene involved in encoding an enzyme that supports the biogenesis of small interfering RNAs and miRNAs, was inactivated in the mouse lung epithelium [12]. There were decreased branching morphogenesis (E12.5 and E13.5) and increased mesenchymal *Fgf10* as well as epithelial *Spry2* and *Bmp4* levels at E12.5 [12]. Following this publication, several lines of evidence reported various levels of miRNAs expression in the lung and their contribution to biological processes during organ development and pathogenesis of some lung diseases.

## 2. Consequences of miRNAs Expression in the Developing Lung

The development of human and mouse lungs are characterized by similar divisions spanning from pseudoglandular, canalicular, terminal saccular, to a final alveolar stage [11]. Regarding miRNAs involvement in lung morphogenesis, we discussed what is known in humans, mice, and rats (see Figure 1). Furthermore, profound genetic similarities have been reported between humans and the two rodents [13,14].

After fertilization in the mouse, total miRNA in the two-cell-stage embryo was demonstrated to be significantly lower than the levels in one-cell zygote [15]. Notably, total miRNA in the four-cell-stage embryo was higher than the levels in the two-cell-stage embryo [15]. In the early mouse embryo, miR-127 was upregulated at E6.5 and E7.5 [16]. Overexpression of miR-127 in mouse embryonic stem cells, which differentiated into embryoid bodies, increased the mRNA levels of *Gsc*, *Foxa2*, and *Brachyury* (mesendoderm markers), but elicited no change in *Pax6* and *Otx2* (ectoderm markers) three days after transfection. Inhibition of miR-127 showed opposite regulation of *Gsc*, *Foxa2*, and *Brachyury*. miR-127, moreover, was suggested to control mesendoderm differentiation [16]. miR-326 regulated sonic hedgehog signaling by targeting *Smo* and *Gli2* [17]. The inhibition of smoothened in cultured explanted E12 mouse lung resulted in the expansion of distal epithelium, and disruption of normal branching pattern and mesenchymal integrity [17]. Another study described the functional role of miR-142-3p in the mouse embryonic lung mesenchyme. The miRNA was shown to regulate adenomatous polyposis coli to further modulate Wnt signaling [18]. Taken together, miR-142-3p regulated the proliferation and differentiation of mesenchymal progenitors in the mouse embryonic lung [18].

In the developing mouse lung, the level of miR-17 was stable (E11.5–E16.5), predominantly at the pseudoglandular stage, with higher epithelial levels at E12.5 [19]. Simultaneous knockdown of miR-17 and its paralogs, miR-20a and miR-106b, for 72 h in epithelial lung explants altered branching, even in FGF10-treated condition [19]. In the human fetal lung, miR-449a was upregulated at 18-20 weeks (canalicular stage), and in the mouse embryonic lung, miR-449a level was increased from E15.5 to E18.5 [20]. The inhibition of miR-449a in E16.5 mouse lung culture (end of pseudoglandular stage) increased the mRNA levels of *Mycn* and *Sox9*, and the protein levels of Ki-67 and SOX9 particularly in the distal epithelial region [20].

Knockout of miR-26a-1/miR-26a-2 in the mouse promoted the formation of dilated lumens and aerated regions at the beginning of the canalicular stage (E16.5) of lung development [21]. Results at the saccular stage (E18.5) also indicated large lamellar bodies, and increased SP-A, SP-B, and SP-C protein levels in miR-26a knockout mice. The study further proposed a role for miR-26a in the synthesis of pulmonary surfactant [21]. The loss of epithelial histone deacetylase 3 in E18.5 mouse lung resulted in increased levels of miR-17-92 cluster and miRNAs in the *Dlk1-Dio3* locus [22]. Overexpression of miR-17-92 in the epithelium of embryonic mouse lung revealed remarkably packed AT1 cells. Together, the study showed that histone deacetylase 3 regulated the level of miR-17-92, which further modulated TGF-β signaling and subsequently, AT1 cell spreading as well as saccular formation [22]. In the fetal rat lung, the highest level of miR-127 was reported at E21 [23]. Furthermore, overexpression of miR-127 in fetal rat lung (E14) culture increased terminal and internal bud sizes.

Regarding alveolarization, several studies have been done in bronchopulmonary dysplasia (BPD) mouse models. For example, hyperoxia-induced impairment in alveolarization was partially mitigated in miR-34a^−/−^ mice [24]. In another study, the administration of miR-29b, nonetheless, improved alveolarization and attenuated fibrotic signaling in postnatal day 28 mice (exposed to 85% O_2_) whose mothers were intraperitoneally administered with LPS during pregnancy [25]. The administration of miR-876-3p mimic to postnatal day 14 mice exposed to hyperoxia restored alveolar structures and reduced neutrophilic inflammation [26]. After hyperoxia exposure, miR-30a-3p and -5p levels increased significantly in female mice lungs on postnatal days 7 and 21 [27]. Similarly, in neonatal human pulmonary microvascular endothelial cells exposed to hyperoxia, samples obtained from females expressed significantly higher levels of miR-30a-3p and -5p compared to levels in male samples [27]. Altogether, there were significant decreases in miR-30a-3p and -5p levels, and increased mRNA and protein levels of DLL4 in human BPD lung samples. [27]. In another set of experiments carried in newborn mice, hyperoxia disrupted alveolar architecture and the injection of an inhibitor for miR-421 attenuated the process [28]. Collectively, miR-421 regulated *Fgf10*, induced apoptosis in the lung, and increased inflammatory response in a BPD mouse model [28]. Details on the expression profile of miRNAs at various stages of late lung development in experimental BPD models have been summarized by Nardiello and Morty [29]. Moreover, the studies discussed so far implicate miRNAs in the regulation of alveolarization.

## 3. The Role of miRNAs in Some Developmental Milestones of the Lung

Besides data on specific epochs of lung development, some studies have reported the expression and role of miRNAs at various levels in the fetal and postnatal lungs. 

A profile of miRNAs in the developing lung revealed an increased level of miR-17-92 cluster at E11.5, which gradually decreased until adult life [30]. Overexpression of miR-17-92 cluster in the lung epithelium of transgenic embryos impaired normal lung development [30]. At E16, E19, and E21 of rat lung development, a wide diversity of miRNAs were regulated [31]. The following miRNAs, let-7a, miR-93, miR-125b-5p, miR-146b, miR-296, miR-1949, and miR-3560 were shown to be consistently regulated in all three stages investigated in the rat embryonic lungs [31]. Between the three stages, miR-1949, miR-125b-5p, miR-296, and miR-93 were downregulated whereas let-7a, miR-146b, and miR-3560 were upregulated. The seven miRNAs were further implicated in rat fetal lung development [31].

During the differentiation of epithelial cells (stimulated with 1mM Bt_2_cAMP) isolated from midgestational human fetal lungs, the level of miR-29 family was significantly elevated [9]. This coincided with increased mRNA level of *SFTPA* in epithelial cell at 48 h and 72 h of culture in the presence of 1mM Bt_2_cAMP. Similar regulatory patterns were observed in epithelial cells isolated from mouse fetal lungs (E15.5–E18.5). Knockdown of miR-29 family promoted TGF-β signaling, which in turn decreased the protein level of SP-A in human fetal lung epithelial cells [9]. The inhibition of miR-200 family in type II cells isolated from human fetal lungs decreased protein levels of SP-A, SP-B and pro-SP-C, and furthermore, reduced phospholipid-containing lamellar inclusions [32]. Similarly, in another study, knockout of miR-200b resulted in the disruption of surfactant properties, decreased lung septation, and thickened alveolar walls [33].

In the developing mouse lung, the relevance of miR-302/367 cluster was demonstrated in endoderm progenitors [34]. Overexpression of miR-302/367 cluster increased the thickness of epithelial lining and reduced sacculation in the embryonic lung (E18.5) [34]. Moreover, in the study, miR-302/367 regulated the proliferation of lung progenitors by repressing *Cdkn1a* and *Rbl2*. The miRNA cluster further modulated lung endodermal progenitor cell polarity by downregulating *Tiam1* and *Lis1* mRNA and protein levels [34].

## 4. miRNAs in Lung Diseases

Clinical evidence has indicated decline in lung function of some adults who survived BPD during early life [35,36]. It is speculative that the resolution of impairments at the alveolar stage of human lung development may leave quiescent imprints that could influence respiratory activities and molecular pathogenesis of lung diseases later in life. We have explained the role of some miRNAs in lung development. Nevertheless, there exists an extensive literature on the regulatory effects of miRNAs in lung diseases. Moreover, the molecular mechanisms that regulate the risk of developing lung diseases in later life, in some neonates/infants who survived respiratory disorders, remain unclear. We propose that the role of miRNAs in signaling pathways that stimulate lung diseases after neonatal/infant life could provide hints for studies intending to bridge these separate mechanisms. Additionally, the epigenetic hallmarks of miRNAs isolate them as one of the classes of molecules that require attention for future studies.

In this review, we focused on the following lung diseases, that is, chronic obstructive pulmonary disease (COPD), asthma, pulmonary fibrosis, and pulmonary arterial hypertension (PAH), because of their impact on global morbidity, mortality, and the common presence of inflammation and tissue remodeling during their pathogenesis. We also considered their pathological associations and availability of experimental models and data, as in the case of PAH (group 1 pulmonary hypertension). It was reported that noncommunicable diseases accounted for 71% of global deaths with chronic respiratory diseases (CRDs) (7%) ranking second to cancers (16%) and third to cardiovascular diseases (31%) [37]. Among the CRDs, COPD and asthma constituted the highest disease burden [38]. Despite obvious pathological differences between asthma and fibrosis, fibroblast accumulation was observed in the proximal airways of patients with severe persistent asthma [39]. Additionally, perichondrial fibrosis was confirmed in the airways of autopsied lungs from patients with bronchial asthma, and was also suggested in the lungs of COPD patients [40]. Asthma exhibits three pathological features with PAH, and this was related to the activation of NFAT in both diseases [41]. Lastly, pulmonary hypertension is common in COPD and idiopathic pulmonary fibrosis [42,43,44].

### 4.1. COPD

COPD is characterized by progressive loss of lung function provoked by chronic bronchitis and emphysema [45]. The pathogenesis of COPD is associated with abnormal inflammatory response in the airways, and recent studies have reported the importance of miRNAs in regulating this process. For example, besides miR-145-5p role in decreasing nuclear translocation of NF-κB p65, the miRNA targeted *KLF5* to inhibit inflammation and apoptosis in cigarette smoke extract (CSE)-treated human bronchial epithelial cells (HBECs) [46]. The transfection of miR-29b mimic into HBE4-E6/E7 cell line downregulated the mRNA and protein levels of BRD4 [47]. miR-29b significantly decreased the mRNA level of IL-8 by regulating *BRD4* in CSE-treated HBE4-E6/E7 cells [47]. In another line of evidence, treatment of CSE to murine monocytic cell line (THP-1) increased the concentrations of IL-1β and TNF-α, protein levels of TLR-4 and NF-κB p65, and decreased the level of miR-149-3p [48]. Transfection of miR-149-3p mimic into CSE-induced THP-1 cells, nonetheless, decreased the level of the proteins mentioned [48]. Bronchial biopsies from COPD patients who were treated with corticosteroids for 6 and 30 months showed increased levels of miR-320d and miR-339-3p, and decreased levels of miR-708 and miR-155 [49]. CSE stimulation of primary bronchial epithelial and BEAS-2B cells overexpressing miR-320d decreased the release of CXCL8 [49]. Additionally, treatment of IL-1β to BEAS-2B cells overexpressing miR-320d downregulated the activation of NF-κB [49]. In CSE-induced human bronchial epithelial (HBE) cells, miR-218 decreased the level of mRNA and release of IL-6 and IL-8 [50]. Similarly, the protein and mRNA levels, as well as the release of MUC5AC by CSE-induced HBE cells were also reduced as a result of miR-218 overexpression. Taken together, miR-218 regulated TNFR1/NF-κB signaling to modulate MUC5AC production and inflammation in CSE-stimulated HBE cells [50]. The inhibition of miR-3202 increased the levels of IFN-γ and TNF-α in a co-culture of T lymphocytes and HBE cells [51]. Furthermore, in a COPD rat model, cigarette smoke (CS) exposure elevated the concentrations of IFN-γ and TNF-α, increased the protein level of FAIM2, disrupted the lung architecture, and downregulated the level of miR-3202 [51]. In another set of experiments, miR-223 was suggested to induce *CX3CL1* mRNA level by regulating histone deacetylase 2 in human pulmonary artery endothelial cells (HPAECs) [52]. In human lung tissues from two cohorts of COPD patients, there were negative correlations between miR-223 and histone deacetylase 2 levels [52]. In another study, miR-181c regulated *CCN1*, which was upregulated in the lung tissues of CS-exposed mice as well as COPD patients [53]. Nevertheless, the inhibition of miR-181c increased the mRNA levels and release of IL-6 and IL-8 in CSE-treated HBECs and CS-exposed mice.

As a form of pre-clinical therapeutic approach, the inhibition of some miRNAs was demonstrated to alleviate inflammatory phenotypes in COPD. In CS-exposed mice, anti-miR-130a decreased the BAL macrophage, neutrophil, dendritic cell, CD4+ T cell, and CD8+ T cell counts [54]. Furthermore, the levels of IL-1β, TNF-α, and IL-6 were reduced in BALF. The inhibition of miR-130a in CSE-treated BEAS-2B cells upregulated the protein levels of WNT1, β-catenin, and LEF1 [54]. In an earlier study, intranasal administration of anti-miR-195 to mice mitigated CS-induced disruption of the lungs [55]. Anti-miR-195 increased the protein level of PHLPP2, and decreased p-AKT in the lungs of CS-exposed mice. Altogether, miR-195 targeted *PHLPP2* and also promoted the release of IL-6 and TNF-α, an observation similar to knockdown of *PHLPP2* in BEAS-2B cells [55].

There have been general reports of miRNA regulation of various molecules in experimental COPD models. In serum samples of COPD patients as well as CSE-treated NCI-H292 cells, miR-212-5p level was shown to be reduced [56]. Nonetheless, miR-212-5p increased p-AKT and downregulated the mRNA and protein levels of IGFBP3. Moreover, opposite observations were reported in CSE-stimulated NCI-H292 cells [56]. The authors could not show the possible role of miR-212-5p in CSE-treated conditions. There is evidence that peripheral blood mononuclear cells and lung tissues obtained from COPD patients exhibited a higher level of miR-664a-3p compared to smokers [57]. Additionally, overexpression of miR-664a-3p in BEAS-2B cells negatively regulated *FHL1* mRNA and protein levels. The authors suggested further work to elucidate the roles of miR-664a-3p and FHL1 in COPD [57].

### 4.2. Asthma

Asthma is marked by recurrent episodes of symptoms (for example, breathlessness, cough and wheezing), airway obstruction, inflammation, and bronchial hyper-responsiveness to various forms of stimuli [58]. The role of miRNAs in asthma-related signaling pathways have been clearly explained by Mousavi *et al.* [59] and Taka *et al.* [60]. Nonetheless, the following are studies that emerged in recent months. In severe asthmatic bronchial epithelial cells obtained from patients, the level of miR-744 was decreased compared to its expression in normal control cells [61]. miR-744 downregulated the mRNA and protein levels of TGFB1, and in TGF-β1-stimulated normal and severe asthmatic bronchial epithelial cells, miR-744 mimic decreased p-SMAD3 and also increased the protein level of SARA. miR-744 decreased the proliferation of bronchial epithelial cells, suggesting the importance of the miRNA in epithelial functions in asthma [61]. miR-943-3p was shown to regulate the level of *SFRP4* [62]. Inhibition of miR-943-3p, however, reduced OVA-induced subepithelial fibrosis and smooth muscle thickness in the mouse lung. miR-943-3p repression of *SFRP4* was reported to support remodeling of the airway in OVA-induced mice [62]. Lung tissues obtained from asthma patients showed increased and decreased protein levels of Beclin-1 and P62, respectively [63]. Furthermore, it was demonstrated that miR-30a regulated *ATG5* to inhibit autophagy-induced airway fibrosis [63].

### 4.3. Pulmonary Fibrosis

Pulmonary fibrosis usually occurs in diseases such as rheumatoid arthritis and systemic sclerosis. Fibrosis of the lung is devastating when there are no established underlying causes, as manifested in idiopathic pulmonary fibrosis (IPF), an interstitial lung disease with poor prognosis and median survival from 2.5 to 3.5 years [64,65]. Until now, experimental models are being developed to recapitulate the progressive pathophysiology of IPF for accurate molecular studies. Nevertheless, the profibrotic impact of transforming growth factor-β (TGF-β) in pulmonary fibrosis has been widely reported in past years. Many *in vitro* and *in vivo* studies have employed TGF-β1 and bleomycin treatments as experimental models for pulmonary fibrosis. In this review, we discussed miRNAs and their regulation of TGF-β signaling, fibroblast differentiation, collagen production, apoptosis, and some phenotypes of pulmonary fibrosis.

Treatments of TGF-β1 and bleomycin downregulated the level of miR-18a-5p, and inhibition of TGF-β1 in bleomycin-stimulated pleural mesothelial cells potentially increased the level of the miRNA [66]. In the study, miR-18a-5p targeted *TGFBR2* and inhibition of miR-18a-5p increased p-SMAD2/3, disrupted mouse lung architecture, and stimulated sub-pleural fibrosis [66]. Congruence to this experimental data, miR-153 was also shown to regulate *TGFBR2* [67]. miR-153 inhibited the phosphorylation of SMAD2/3 and consequently, decreased the protein levels of fibronectin and α-SMA. Thus, the miRNA regulated fibrogenesis in the human pulmonary fibroblast line MRC-5 [67]. Similarly, miR-1343 suppressed *TGFBR1* and *TGFBR2*, and also reduced the levels of markers of fibrosis (collagen type I and α-SMA) in TGF-β1-stimulated primary lung fibroblasts [68]. In another study, the mRNA and protein levels of TGFBR2 and NOX4, as well as p-SMAD2 level, were decreased by miR-9-5p in human fetal lung fibroblasts (HFL-1) [69]. Similar observation was reported in the mouse. Orotracheal administration of bleomycin following lenti-miR-9 treatment interrupted structural damage to the lungs, hence prevented pulmonary fibrosis in mice [69]. miR-101 decreased the mRNA and protein levels of TGFBR1, FZD4 and FZD6, and thus blocked NFATc2 and TGF-β1 signaling to attenuate fibroblast proliferation and activation [70].

In an *in vitro* model of pulmonary fibrosis, miR-486-5p mimic was shown to inhibit TGF-β1 activation of SMAD2 and further, decreased the protein levels of fibronectin, α-SMA and CTGF in mouse fibroblasts (NIH/3T3) [71]. Intratracheal administration of miR-486-5p in a lung fibrosis mouse model also decreased myofibroblast differentiation and collagen deposition. Besides *SMAD2* binding, miR-323a-3p also targeted *TGFA* [72]. miR-323a-3p mimic repressed the mRNA and protein levels of SMAD2 and TGFA in human bronchial epithelial cells (16HBE14o-). miR-323a-3p also downregulated Caspase 3 to reduce the apoptosis of 16HBE14o- cells. Taken together, miR-323a-3p inhibited bleomycin-induced lung fibrosis in mice [72].

During the pathogenesis of pulmonary fibrosis, TGF-β1 usually stimulates the transdifferentiation of fibroblasts to myofibroblasts. Conversely, as shown in a study, miR-27a-3p inhibited TGF-β1 activation of human lung fibroblasts (MRC-5 line) [73]. It was suggested in another study that miR-489 targeted *Myd88* to decrease protein levels of IL-1β and TGF-β1 in silica-induced lung fibrosis mouse model [74]. miR-489 attenuated fibroblast differentiation by blocking the phosphorylation of SMAD3 in TGF-β1-stimulated NIH/3T3 cells [74]. Similarly, miR-133a downregulated the protein level of CTGF, and the mRNA levels of *COL1A1* and *ACTA2* to inhibit pulmonary fibroblast differentiation and to prevent fibrosis phenotype in TGF-β1-treated conditions [75]. Another line of evidence reported that miR-542-5p regulated the mRNA and protein levels of ITGA6 in TGF-β1-stimulated NIH-3T3 cells [76]. Knockdown of *Itga6* in TGF-β1-treated NIH-3T3 cells nonetheless, inhibited markers of fibrosis and also blocked the phosphorylation of FAK and AKT [76]. miR-542-5p was suggested to regulate *Itga6* level, and thus attenuated fibroblast differentiation via the FAK signaling pathway.

In pulmonary fibrosis, the regulation of collagen production is important for the therapeutic treatment of the disease. miR-29a and miR-29c were demonstrated as examples of miRNAs that regulate collagen production in pulmonary fibrosis. miR-29a downregulated the mRNA and protein levels of LOXL2 and SERPINH1 in MRC-5 fibroblasts [77]. It is noteworthy that *LOXL2* and *SERPINH1* genes are involved in the synthesis of collagen. Therefore, miR-29a elicits anti-fibrotic properties. Knockdown of histone deacetylase C4 resulted in an increase in type I collagen and a decrease in miR-29c level in control human lung fibroblasts [78]. Additionally, in IPF fibroblasts overexpressing protein phosphatase 2A, the silencing of histone deacetylase C4 increased type I collagen and decreased the level of miR-29c [78].

Recent developments have shown the potential roles of miRNAs in the regulation of apoptosis in pulmonary fibrosis. A study demonstrated that TGF-β1 downregulated miR-29c and the protein level of cell surface death receptor, Fas, in HFL-1 cells [79]. miR-29c mimic however upregulated the protein abundance of Fas and also induced apoptosis in TGF-β1-stimulated HFL-1 cells [79]. Following this publication, another study suggested that knockout of mir-29c in mouse alveolar epithelial type II cells induced apoptosis and decreased cell renewal [80]. miR-29c regulated the level of *Foxo3a* to inhibit the apoptosis of alveolar epithelial type II cells. Overexpression of miR-29c in alveolar epithelial type II cells decreased the level of hydroxyproline in a bleomycin-induced lung fibrosis mouse model [80]. miR-34a^−/−^ mice exhibited severe fibrosis phenotype after bleomycin instillation [81]. It was further revealed that primary lung fibroblasts isolated from bleomycin-treated miR-34a^−/−^ mice became less senescent. Additionally, there was decreased cell apoptosis in the lungs of bleomycin treated miR-34a^−/−^ mice [81].

In most studies, the regulatory effects of miRNAs in pulmonary fibrosis are demonstrated to target the alleviation of the disease. For example, miR-708-3p mimic regulated *ADAM17* and also downregulated markers of fibrosis in TGF-β1-treated MRC-5 cells [82]. Administration of synthesized miR-708-3p agomir to mice treated with bleomycin resulted in thinner alveolar walls [82]. In another study, miR-338-5p attenuated bleomycin-induced fibrosis in mice [83]. Similarly, miR-338-5p regulated LPA1 protein level to decrease fibrosis phenotype in bleomycin-treated mouse lungs [84].

### 4.4. PAH

PAH is one of five classifications of pulmonary hypertension [85], and leads to right ventricular failure as well as premature death. One major pathological feature of PAH is vascular remodeling. There are a growing number of publications regarding the role of miRNAs in some categories of pulmonary hypertension. Jusic *et al.* [86] have recently reviewed what is already known in pulmonary arterial hypertension. However, since their publication, there is new evidence to expand knowledge in this field of active research.

Some studies have demonstrated that miRNAs contribute to the alleviation of PAH. For instance, the proliferation of human pulmonary arterial smooth muscle cells (hPASMC) was inhibited by miR-205-5p [87]. It was further explained that miR-205-5p regulated MICAL2, a protein that proliferated hPASMCs via the activation of ERK1/2 signaling [87]. The mRNA level of *FGF-7* was upregulated in tissues obtained from PAH patients and in PAH-PASMCs [88]. miR-455-3p-1 alleviated PAH by regulating *FGF7* and RAS/ERK signaling, respectively [88]. In a recent study, there were decreased levels of miR-181a-5p and miR-181b-5p, whereas the protein level of endocan was elevated in monocrotaline-induced PAH in a time-dependent manner [89]. miR-181a/b-5p decreased the protein level of endocan to mitigate inflammatory response in TNF-α-treated primary rat pulmonary arterial endothelial cells [89].

There is compelling evidence that some miRNAs promote the pathogenesis of PAH. It was recently shown that hypoxia exposure stimulated PAH and also increased miR-27a level in the rat pulmonary artery and in HPAECs [90]. Moreover, miR-27a targeted *SMAD5* to promote hypoxia-induced endothelial to mesenchymal transition [90]. Chronic normobaric hypoxia stimulated PAH and also increased the level of miR-335-3p in the mouse lung [91]. The inhibition of miR-335-3p resulted in the attenuation of chronic normobaric hypoxia-induced vascular remodeling and PAH in mice [91]. A significant increase in miR-30a-5p level in the plasma of PAH patients was recently reported [92]. Hypoxia exposure to HPAECs progressively downregulated miR-30a-5p and upregulated YKL-40 levels in a time-dependent manner. Furthermore, it was evidently indicated that miR-30a-5p targets *YKL-40*. To investigate the functional role of the miRNA, HPAECs were exposed to hypoxia after transfection with miR-30a-5p mimic. Consequently, miR-30a-5p enhanced the proliferation and attenuated the apoptosis of HPAECs [92].

Figure 2 shows a summary of miRNAs involved in pulmonary arterial hypertension and the other lung diseases discussed. We have also summarized the miRNAs and their target genes in lung development and diseases (Table 1).

## 5. Comparative Role of miRNAs in Lung Development and Diseases

Some review articles have mentioned the involvements of TGF-β/BMP, Wnt, sonic hedgehog, and FGF10 signaling in lung development and diseases [93,94,95,96,97]. The signaling pathways of these molecules are regulated by miRNAs. Considering the role of miRNAs in lung development and the pathogenesis of some diseases, it is crucial to identify specific miRNAs and their regulatory influence in the two biological processes.

We observed a dual function of miR-34a, specifically in the late stage of lung development and pathogenesis of IPF. In newborn mice, miR-34a impaired alveolarization, and in adult mice, the miRNA alleviated the progression of pulmonary fibrosis [24,81]. Figure 3 illustrates the functions of miR-34a in the mouse lung.

Furthermore, depending on targeting sites and the pathogenesis of the disease, some categories of miRNAs may exhibit various regulatory functions in the lung. For example, miR-27a-3p attenuated pulmonary fibrosis, and in another report, miR-27a supported PAH [73,90]. Similarly, miR-30a interrupted the development of asthma, and on the other hand, miR-30a-5p promoted PAH [63,92]. miR-29a and miR-29c inhibited the pathogenesis of IPF, and miR-29b regulated inflammation in COPD [47,77,79,80]. miR-181c attenuated the pathogenesis of COPD and miR-181a/b-5p were involved in the regulation of PAH [53,89].

## 6. Perspective

As a recommendation, miRNAs are receiving remarkable attention to be used as biomarkers and clinical diagnosis of lung diseases. For example, recent attempts to find biomarkers in the serum samples of children suffering from asthma revealed associations between some miRNAs and functional metrics of the lungs [98]. Among the numerous circulating miRNAs detected in serum samples of research participants, 22 were associated with FEV_1_/FVC, eight with FVC%, and four with FEV_1_% [98]. Apart from asthma, the concept of using miRNAs as biomarkers in COPD was recently described by Salimian *et al.* [99]. Other lung diseases such as sarcoidosis and cancer have also been connected with numerous miRNAs [100]. Conversely, the circulating miRNAs in patients with lung diseases may only indicate the presence of the conditions and not necessarily their progression. We suggest the screening of miRNAs at various pathophysiological stages of pulmonary diseases to address this concern. Moreover, to ensure healthy human lung, further work is required to standardize miRNAs for the diagnosis of lung diseases.

Beyond their utility as biomarkers, miRNAs are important to understand the mechanistic underpinnings of disease pathophysiology in the context of lung development. Thus, miRNAs are essential for lung development with particular interest in the alveolar stage, which has notable examples in BPD patients. In a recent study, intravenous administration of miR-302b/c mimic after *Streptococcus pneumoniae* infection in mice proliferated AT1 and AT2 cells, and also increased survival [10]. let-7d was transduced into bone marrow-derived mesenchymal stem cells (let-7d), which were intravenously injected into bleomycin-treated mice [101]. This caused decreased inflammatory cells in the lungs, and mice began to recover from weight loss induced by bleomycin instillation two days after let-7d treatment [101]. Reports on the role of miRNAs in the alveolar stage of lung development could support efforts made to promote normal tissue repair and regeneration after injury. We also propose the characterization of miRNAs expressed in stem cells and progenitor cells of the lung to support the programming of regeneration and restoration of normal organ function after severe tissue damage.

In summary, the prospects of acquiring lung diseases in early life have been partly associated with molecular factors *in utero*. The expression of miRNAs at various stages of lung development and diseases widely project their contribution to human lung health. We speculate that the involvement of miRNAs in the pseudoglandular and canalicular stages of lung development suggests their relevance in reepithelialisation in some human lung diseases. Additionally, the influence of miRNAs in the late stage of lung development could enlighten efforts being made toward alveolar repair in lung diseases.

## Figures and Tables

**Figure 1 ijms-21-02765-f001:**
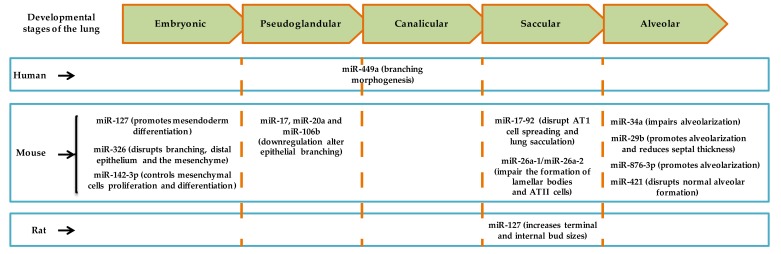
Summary of the role of miRNAs at various stages of human, mouse, and rat lung development as indicated in cited articles.

**Figure 2 ijms-21-02765-f002:**
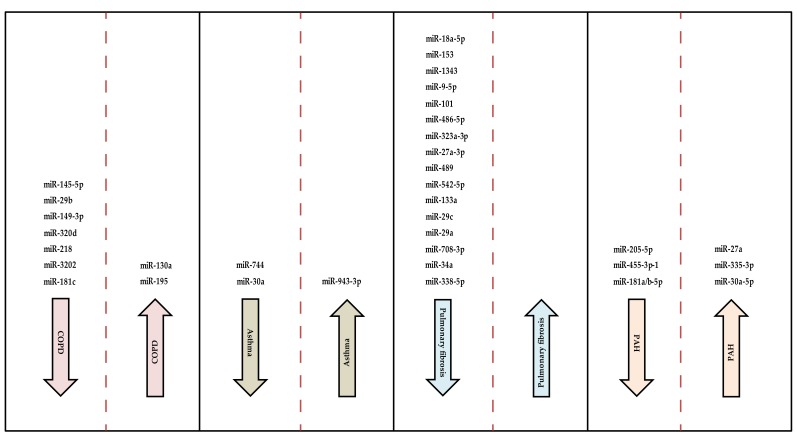
Schematic representation of miRNAs in pulmonary diseases as already cited from published articles. Arrows pointing downwards indicate the inhibition of the pathogenesis of the diseases by the miRNAs in the column. miRNAs in columns with arrows pointing upwards progress the development of the diseases. No miRNA was found to be involved in the progression of pulmonary fibrosis in this review.

**Figure 3 ijms-21-02765-f003:**
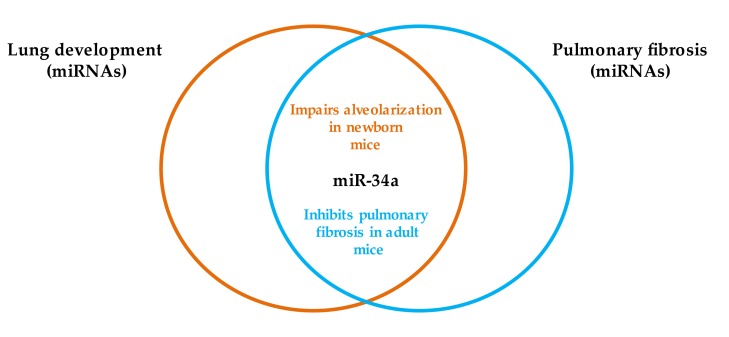
miR-34a regulates lung development and the pathogenesis of pulmonary fibrosis.

**Table 1 ijms-21-02765-t001:** miRNAs and their target genes in lung development and diseases, as shown in cited articles.

miRNA	Molecular Target(s)
**Lung Development**	
miR-326	*Smo and Gli2*
miR-142-3p	*Apc*
miR-449a	*MYCN*
miR-421	*Fgf10*
miR-302/367	*Cdkn1a*, *RbI2*, *Tiam1,* and *Lis1*
**Lung Diseases**	
**COPD**	
miR-145-5p	*KLF5*
miR-149-3p	*TLR-4*
miR-218	*TNFR1*
miR-3202	*FAIM2*
miR-223	*HDAC2*
miR-181c	*CCN1*
miR-195	*PHLPP2*
miR-664a-3p	*FHL1*
**Asthma**	
miR-744	*TGFB1*
miR-943-3p	*SFRP4*
miR-30a	*ATG5*
**Pulmonary Fibrosis**	
miR-18a-5p	*TGFBR2*
miR-153	*TGFBR2*
miR-1343	*TGFBR1* and *TGFBR2*
miR-9-5p	*TGFBR2* and *NOX4*
miR-101	*TGFBR1*, *FZD4,* and *FZD6*
miR-486-5p	*SMAD2*
miR-323a-3p	*SMAD2* and *TGFA*
miR-489	*Smad3*
miR-133a	*TGFBR1*, *CTGF,* and *Col1a1*
miR-542-5p	*Itga6*
miR-29a	*LOXL2* and *SERPINH1*
miR-708-3p	*ADAM17*
miR-338-5p	*LPA1*
**PAH**	
miR-205-5p	*MICAL2*
miR-455-3p-1	*FGF7*
miR-181a/b-5p	endocan
miR-27a	*Smad5*
miR-30a-5p	*YKL-40*

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
