# Peer review of "miRNAs in Lung Development and Diseases"

_ijms, 2020, doi:10.3390/ijms21082765_

Round 1

Reviewer 1 Report

The manuscript “miRNAs in lung development and diseases” by Eistine Boateng and Susanne Krauss-Etschmann is a review article focused on the role of microRNAs in the regulation of lung development and diseases.

This manuscript is rich of details and relevant to readers involved in microRNA research in the lung development field.

The authors have researched topics thoroughly; nonetheless I have few suggestions which might be taken in consideration:

- Inclusion of a table summarising the main gene target and/or molecular pathways of some relevant miRNA involved in lung development and diseases should be helpful.

- Albeit, as clearly stated in the title, the review is focused on miRNAs, authors should consider to briefly mentioning the role of other non-coding RNAs.

Minor issues:

- Page 2, Line 48: “thus, suggested….”: revise

- Page 2, line 65: “knockout”; capitalize k

- Page 4, line 158: “Pre-treated lenti-miR-9”: explain

- Page 5, line 185;” Considering the molecular targets of miRNAs, some successful attempts have been made so far.” Explain

- Page 5 line 188 “ hence, suggests miR-29a to mediate anti-fibrotic responses.” Rephrase

- Page 5 Line 210 “Figure 2 shows a summary of the role of miRNAs in pulmonary fibrosis”. Figure 2 does not show any “role of miRNAs”. The same on page 7 line 303

Author Response

We have included a table (Table 1) as suggested. The title of the table is “Table 1. miRNAs and their target genes in lung development and diseases as indicated in cited articles” On page 8, line 324

On page 7, line 316, we introduced the table as followsWe have also summarized the miRNAs and their target genes in lung development and diseases (Table 1)

- Albeit, as clearly stated in the title, the review is focused on miRNAs, authors should consider to briefly mentioning the role of other non-coding RNAs.

We respectfully agree with this suggestion as there are emerging roles of some of these non-coding RNAs example, LncRNAs in lung cancer, pulmonary fibrosis and asthma. However, we wish to focus our attention on miRNAs for now. We will consider writing about the others in the future.

Minor issues:

- Page 2, Line 48: “thus, suggested….”: revise

Replaced with “miR-127 was suggested to control mesendoderm differentiation’’ On page 2, line 49

- Page 2, line 65: “knockout”; capitalize k

Corrected on page 2, line 66

- Page 4, line 158: “Pre-treated lenti-miR-9”: explain

Pre-treated lenti-miR-9 to mice repressed bleomycin-induced structural damage to the lungs hence, prevented pulmonary fibrosis.”

Replaced with

“Orotracheal administration of bleomycin following lenti-miR-9 treatment interrupted structural damage to the lungs hence, prevented pulmonary fibrosis in mice”. Now page 4, line 164

- Page 5, line 185;” Considering the molecular targets of miRNAs, some successful attempts have been made so far.” Explain

Considering the molecular targets of miRNAs, some successful attempts have been made so far. For example, miR-29a downregulated the mRNA….

Deleted and replaced with

“Interestingly, miR-29a and miR-29c were demonstrated as examples of miRNAs that regulate collagen production in pulmonary fibrosis. miR29a downregulated …..”. Now page 5, line 191

- Page 5 line 188 “ hence, suggests miR-29a to mediate anti-fibrotic responses.” Rephrase

hence, suggests miR-29a to mediate anti-fibrotic responses.”

Replaced with                                                  

“Therefore, miR-29a elicits anti-fibrotic properties. Now page 5, line 194

- Page 5 Line 210 “Figure 2 shows a summary of the role of miRNAs in pulmonary fibrosis”. Figure 2 does not show any “role of miRNAs”.

“Figure 2 shows a summary of the role of miRNAs in pulmonary fibrosis

This line has been deleted

The same on page 7 line 303

“Figure 2 shows a summary of the role of miRNAs in pulmonary arterial hypertension and the……

Replaced with

Figure 2 shows a summary of miRNAs involved in pulmonary arterial hypertension and the………” Now page 7, line 315.

Reviewer 2 Report

In the evaluated review article (ijms-765120), the authors present the main results of the numerous articles related to the role of various miRNAs in subsequent steps of lung development and different lung diseases (excluding lung cancer). Although the lecture of the manuscript is painful it well and in a systematic way summarizes the state of the art in the field. Figure 1 and Figure 2 are doing a very good job, help to keep track of the text. The references are up to date and adequate. I have only a few minor comments that may help to improve the manuscript.

  • I know that it is commonly used jargon but in phrases like in line 62, line 84, and many others, I would recommend replacing the term “expression” with the term ‘level’. Genes are expressed, and the result of gene expression is a higher or lower level of miRNA, mRNA, protein or subsequent metabolite or phenotype.
  • Line 38, phrase “in mice and humans” Shouldn't also be here ‘rat’?
  • Line 29 in mouse the standard ID for the gene is Dicer1 (not Dicer), in human DICER1
  • From most miRNA precursors both miR-5p and miR-3p are generated, but even if one miR is predominantly generated I would recommend (if possible) adding suffix 5p or 3p. Otherwise, it may be misleading for the readers that are not very familiar with this particular miR. Furthermore, even if one miR is predominant in most tissues, still in some tissues or some conditions the miR from the other side may be generated and may play a role. Nomenclatures of miRNAs are constantly changing.
  • I would suggest starting each subchapter dedicated to a particular disease with a short description of the disease. Just basic facts, a short paragraph or a few sentences.
  • Line 167: it is not necessary to indicate that the miR target a 3’UTR, it is a typical location of targeted sequences.
  • All gene IDs should be italicized, please doublecheck the text, e.g, line 180.
  • The authors do not review the role of miRNAs in lung cancer. It is probably a good idea because the role of miRNAs in cancer is a quite extensive and separate subject, reviewed in separate articles. But the fact of lung cancer exclusion should be indicated, if not in the title, at least in the abstract. It would also be good to shortly explain the exclusion of lung cancer in the text, directing the readers interested in lung cancer to one or a few articles/reviews that generally cover the subject. In this context mentioning the lung cancer in the perspective section, line 337 is out of the blue, completely not necessary.
  • Typos: in line 39 is “figure”, should be Figure; in line 65 is “knockout”, should be ‘Knockout’;

Author Response

  • I know that it is commonly used jargon but in phrases like in line 62, line 84, and many others, I would recommend replacing the term “expression” with the term ‘level’. Genes are expressed, and the result of gene expression is a higher or lower level of miRNA, mRNA, protein or subsequent metabolite or phenotype.

We have corrected them in the manuscript as recommended.

                   Example, on page 2:

                                  Line 45, the word “expression” was deleted

                                  Line 57, the word “expression” was replaced with “level”

                                   Line 61, the word “expression” was replaced with “level”

  • Line 38, phrase “in mice and humans” Shouldn't also be here ‘rat’?

We have included ‘rats’ as recommended, Page 1, line 39

We have also rephrased the next sentence,Furthermore, profound genetic similarities have been reported between humans and the two rodents Page 1, line 40

  • Line 29 in mouse the standard ID for the gene is Dicer1 (not Dicer), in human DICER1

We have corrected as recommended, Page 1, line 30

  • From most miRNA precursors both miR-5p and miR-3p are generated, but even if one miR is predominantly generated I would recommend (if possible) adding suffix 5p or 3p. Otherwise, it may be misleading for the readers that are not very familiar with this particular miR. Furthermore, even if one miR is predominant in most tissues, still in some tissues or some conditions the miR from the other side may be generated and may play a role. Nomenclatures of miRNAs are constantly changing.

We agree with this very important comment. However, for all articles referenced in this manuscript, suffixes 5p or 3p were added if indicated in the published papers. We did not want to commit ourselves by either giving annotations of 3p or 5p when authors could not clearly state which one they had investigated. We understand this to be crucial limitation in the scientific community.

  • I would suggest starting each subchapter dedicated to a particular disease with a short description of the disease. Just basic facts, a short paragraph or a few sentences.

We have included as recommended

Pulmonary fibrosis, Page 4, line 144

COPD, page 5, line 218

Asthma, page 6, line 272

Pulmonary arterial hypertension, page 7, line 289

  • Line 167: it is not necessary to indicate that the miR target a 3’UTR, it is a typical location of targeted sequences.

3’ UTR has been deleted from the sentence, Now page 4, line 172

  • All gene IDs should be italicized, please doublecheck the text, e.g, line 180.

We appreciate this recommendation. All have been doublechecked and those identified, corrected accordingly.

Example: Page 5, line 186

  • The authors do not review the role of miRNAs in lung cancer. It is probably a good idea because the role of miRNAs in cancer is a quite extensive and separate subject, reviewed in separate articles. But the fact of lung cancer exclusion should be indicated, if not in the title, at least in the abstract. It would also be good to shortly explain the exclusion of lung cancer in the text, directing the readers interested in lung cancer to one or a few articles/reviews that generally cover the subject. In this context mentioning the lung cancer in the perspective section, line 337 is out of the blue, completely not necessary.

We have stated the lung diseases discussed in the abstract, page 1, line 14.

We followed up with a sentence to inform readers on page 4, line 140

We completely agree with the reviewer and have deleted the reference on lung cancer.

  • Typos: in line 39 is “figure”, should be Figure; in line 65 is “knockout”, should be ‘Knockout’;

We have corrected as recommended.

“Figure”, Page 1, line 40

“Knockout”, Page, line 66
